# Biomechanics of subtrochanteric fracture fixation using short cephalomedullary nails: A finite element analysis

**Dae-Kyung Kwak[1]◉, Sun-Hee Bang[2]◉, Won-Hyeon Kim[3], Sung-Jae Lee[2], Seunghun Lee[1], Je-Hyun Yoo[1]***

**1** Department of Orthopaedic Surgery, Hallym University Sacred Heart Hospital, Hallym University School of Medicine, Anyang, South Korea, **2** Department of Biomedical Engineering, Inje University, Gimhae, South Korea, **3** Department of Mechanical Engineering, Sejong University, Seoul, South Korea

◉ These authors contributed equally to this work.
* oships@hallym.ac.kr

**Data Availability Statement:** All relevant data are within the manuscript and its Supporting information files.

## Abstract

A finite element analysis was performed to evaluate the stresses around nails and cortical bones in subtrochanteric (ST) fracture models fixed using short cephalomedullary nails (CMNs). A total 96 finite element models (FEMs) were simulated on a transverse ST fracture at eight levels with three different fracture gaps and two different distal locking screw configurations in both normal and osteoporotic bone. All FEMs were fixed using CMNs 200 mm in length. Two distal locking screws showed a wider safe range than 1 distal screw in both normal and osteoporotic bone at fracture gaps $\leq$ 3 mm. In normal bone FEMs fixed even with two distal locking screws, peak von Mises stresses (PVMSs) in cortical bone and nail constructs reached or exceeded 90% of the yield strength at fracture levels 50 mm and 0 and 50 mm, respectively, at all fracture gaps. In osteoporotic bone FEMs, PVMSs in cortical bone and nail constructs reached or exceeded 90% of the yield strength at fracture levels 50 mm and 0 and 50 mm, respectively, at a 1-mm fracture gap. However, at fracture gaps $\geq$ 2 mm, PVMSs in cortical bone reached or exceeded 90% of the yield strength at fracture levels $\geq$ 35 mm. PVMSs in nail showed the same results as 1-mm fracture gaps. PVMSs increased and safe range reduced, as the fracture gap increased. Short CMNs (200 mm in length) with two distal screws may be considered suitable for the fixation of ST transverse fractures at fracture levels 10 to 40 mm below the lesser trochanter in normal bone and 10 to 30 mm in osteoporotic bone, respectively, under the assumptions of anatomical reduction at fracture gap $\leq$ 3 mm. However, the fracture gap should be shortened to the minimum to reduce the risk of refracture and fixation failure, especially in osteoporotic fractures.

## Introduction

Osteosynthesis of subtrochanteric fractures is challenging due to the displacement of bone fragments by muscle forces. Intense medial compression and lateral tensile forces are concentrated in the fracture region. These deforming forces make it difficult to achieve anatomical

**Funding:** The authors received no specific funding for this work.

**Competing interests:** The authors have declared that no competing interests exist.

reduction and fixation, often leading to non-union, malunion, and mechanical failures [1,2]. The overall incidence of non-union or delayed union of subtrochanteric fractures and subsequent failure varies from 7% to 20% [3,4].

Cephalomedullary nails (CMNs), due to their biomechanical superiority to extramedullary implants, have been used as the preferred devices for treating subtrochanteric fractures [5,6]. Favorable clinical results have been reported following the treatment of subtrochanteric fractures using CMNs [7,8], and both short and long CMNs are currently used in the management of subtrochanteric fractures. A long CMN has a biomechanical advantage over a short CMN as it provides improved stability due to a longer working length and protects the remnants of the femur shaft below the fracture site [9,10]. Short CMNs have several advantages such as shorter operative and fluoroscopy times, less blood loss, and lower cost than long CMNs [11,12]. Subtrochanteric fractures usually occur in elderly patients, and the incidence of subtrochanteric fractures in osteoporotic elderly patients, including atypical subtrochanteric fractures, is expected to increase in the future [13,14]. It is important to reduce operation time and decrease blood loss during hip fracture surgery in elderly patients with multiple comorbidities and poor physical conditions [15]. Considering these factors, the use of short CMNs would be more advantageous than the use of long CMNs, especially in elderly patients with subtrochanteric fractures. Although CMNs are commonly used for the surgical treatment of subtrochanteric fractures, the indications for the use of short versus long CMNs still remain unclear [11,16,17]. Furthermore, there is little information on appropriate criteria or indications for the usage of short nails in subtrochanteric fractures with various fracture levels and gaps.

Therefore, we conducted this study to investigate the stress in the CMNs and the circumferential cortical bone at various fracture levels and gaps using short nails with two different numbers of distal locking screws in subtrochanteric fracture models and the fracture patterns in which short nails can be used in normal and osteoporotic bones by finite element analysis. Our hypotheses were that 1) short nails can be used limitedly in subtrochanteric fractures; 2) two distal locking screws would have the wider safe range than one distal locking screw; 3) the safe range would be different according to the fracture level and gap, and bone quality.

## Materials & methods

### Finite element model (FEM)

A three-dimensional femoral FEM, verified in previous literatures, was used in this study [18–20]. Computed tomography (CT)-scanning of a left intact femur was performed at 1.0-mm transverse resolution in 1.0-mm increments. After extracting the outline of each CT slice image through reconstruction using the Mimics (version 21.0, Materialise, Leuven, Belgium) program, it was stacked in three dimensions to obtain the line and surface of the entire femoral shape. Lines and surfaces constructed in three dimensions were corrected for distorted areas and then subjected to a segmentation process to obtain a final three-dimensional femoral shape. The volume of the cortical and cancellous bone was created using this shape, and a femoral FEM was implemented through meshing process. To verify the finite element model, strain was measured by attaching a strain gauge at a total of 20 points on the anterior, posterior, medial, and lateral sides of the model, and compared with the previous study according to the method conducted by Heiner et al. [19]. An osteoporotic FEM was reproduced according to the previously verified method [21]. Kose et al. [21] reported that cortical thickness index (CTI) was significant measurement indicators to represent the osteoporotic bone model. CTI is calculated as the ratio of cortical thickness to bone diameter 10 cm distal to the lesser trochanter. At CTI value less than 0.3, the correlation with osteoporotic bone showed 100% sensitivity and 98% specificity. Accordingly, we reproduced the osteoporotic bone model

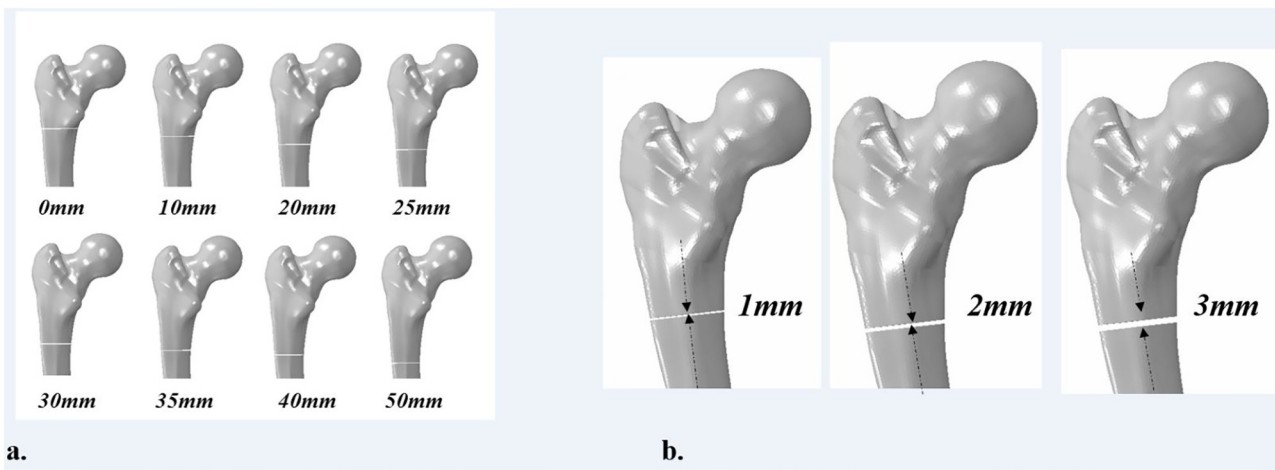

**Fig 1. Finite element models with transverse subtrochanteric fracture of different fracture levels.**

based on this CTI in this study. To reproduce subtrochanteric fracture models, transverse fracture lines at eight levels in the subtrochanteric region [0, 10, 20, 25, 30, 35, 40, and 50 mm below the lower margin of the lesser trochanter and three different fracture gaps (1, 2, and 3mm) for each of the eight sites were produced on each normal and osteoporotic bone model using ABAQUS (version 6.14, Dassault Systems, Paris, France) (Fig 1).

Short Gamma 3 CMNs (Stryker, Mahwah, NJ, USA), 200 mm in length, 12.0 mm in nail diameter with 125° caput-collum-diaphyseal angle, 108-mm lag screw length, and 40-mm distal locking screw length were used in this study. The geometry of all nail configurations was constructed in the intramedullary canal of each FEM using ABAQUS. The lag screw was inserted into the center-to-center position in the femoral head, and the tip-apex distance was set to 22.92 mm in summation of the anterior-posterior and lateral views. Anatomical reduction was assumed in all the models while maintaining the fracture gap. The fracture gap was defined as the longitudinal distance between the two fractured ends at the cortex and the fracture site was assumed free of any bony contact. Short CMNs with two different distal locking screw configurations (1 and 2) were inserted in each FEM using ABAQUS. A total of 96 models were reproduced.

Eight-noded hexahedral elements (C3D8) and four-noded tetrahedral elements (C3D4) were created by using the automatic solid and mesh generation program (ABAQUS/Standard) to build up the mesh of the fractured femur model and the Gamma 3 CMNs. These elements enabled the definition of the different material properties and maintained contact conditions in the fracture plane.

## Material properties

The finite element analysis assumed that the bone structure has two different material properties (cortical bone and density-based homogeneous cancellous bone) and isotropic linear properties. To assign cancellous bone properties to the femoral model, the elastic modulus was calculated based on the referred average CT Hounsfield unit (HU) value of 120.8 [22]. The following bone density-HU and elastic modulus-bone density relationships were used [23,24]:

$$\rho \ = \ 131000 + 1067 \ \mathrm{HU}$$

$$\mathrm{E} \ = \ 6850 \ \rho^{1.49}$$

**Table 1. Material properties applied for the finite element model analysis.**

| | | Elastic Modulus(E) (MPa) | Poisson's ratio(v) |
|---|---|---|---|
| Cortical bone | | 17000 | 0.3 |
| Cancellous bone | Normal bone | 920 | 0.2 |
| | Osteoporotic bone | 574 | 0.2 |
| Implant (TI6Al4V) | | 113800 | 0.3 |

where ρ is the apparent density (g/cm$^3$), and E is the elastic modulus (MPa). The material properties of the femoral cortical bone and nail were referenced from earlier publications (Table 1) [25,26]. Titanium alloy (TI6Al4V) was used for the Gamma 3 CMNs for the purpose of analysis. Different material properties were assigned to different femoral regions.

### Boundary and loading conditions

Assuming the 1-leg stance is taken during normal ambulation, a hip joint force (2013.9 N, 300% of the body weight) was loaded on the femoral head, and an abductor muscle force (671.3 N, 100% of the body weight) was applied to the lateral surface of the greater trochanter [27]. Each force was acting at an angle of 20˚ from the vertical line in the frontal plane (Fig 2).

A "tie" contact condition was applied in this study, assuming full constraints between bone and bone, bone and lag screw, and bone and distal locking screw. The general contact condition was applied using a friction coefficient of 0.42 to allow for optimal movement [28].

A total of 96 FEMs were tested using combinations of eight different fracture levels, three different fracture gap sizes, two different bone qualities, and two different distal locking screw configurations. The stresses in the CMNs and the surrounding cortical bone were investigated with emphasis on the fracture level and gap, and the number of distal locking screws in the FEMs and compared to the yield strength. The yield strength values of cortical bones and CMNs were referenced from earlier publications (Cortical bone, 107.9 MPa; TI6A14V, 880 MPa) [29,30].

### Validation of the FEM with an implanted CMN

To validate the FEM, we reconstructed a FEM and made an analysis to compare with the published experimental data [31]. In the literature, a mechanical experiment was performed using a composite synthetic bone, and a FEM was reproduced according to the experimental model. The FEM with a fracture level of 0 mm below the lesser trochanter and 1 mm fracture gap, which is the same conditions as in the reference literature, was compared and verified by applying the same load. The results were compared through strain values at the anterior and posterior portion of the lag screw hole, and lateral side of the nail. According to the experiment by Eberle et al. [31], the error rate of the strain difference between the experimental model and the FEM with implanted CMN, was 23%. We compared FEM in this study and the above experimental model as the same method; As a result, the error rate between the FEM in this study and the experimental model in the literature was only 9%. Considering these results, the FEM in this study was satisfactorily validated.

## Results

### Stress distribution in cortical bone and nail constructs in FEMs with 1-mm fracture gap

Peak von Mises stress (PVMS) in the cortical bone was observed around the distal locking screws in all FEMs regardless of the fracture level, bone quality, and the number of distal locking screws. The PVMS site on the nail constructs tended to move downwards from the

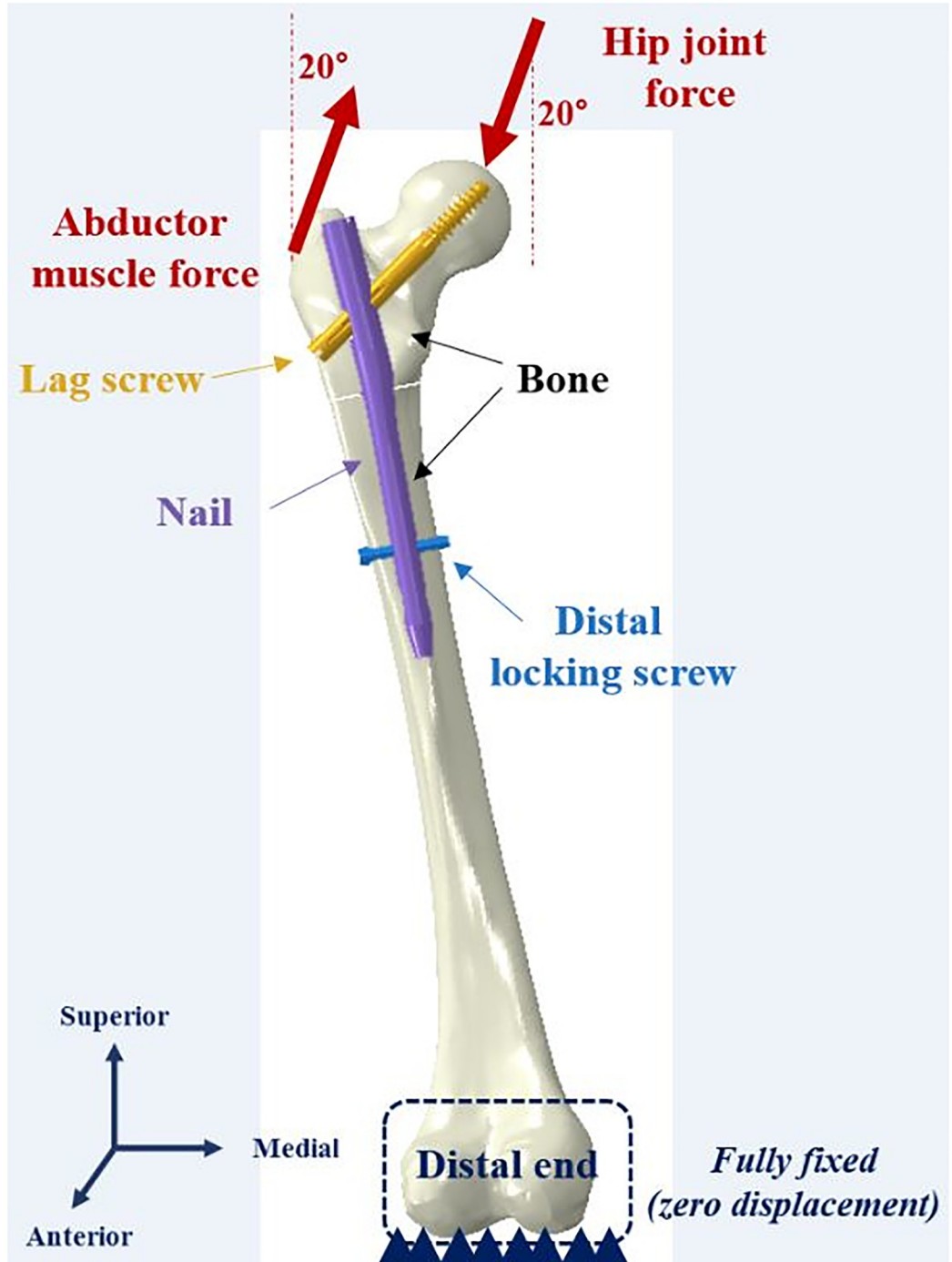

**Fig 2. Loading condition of the analysis model; Hip joint force (FH), 2013.9 N (body weight X 300%); Abductor muscle force, 671.3 N (body weight X 100%).**

junction with the lag screw through the fracture site to the junction with the distal locking screw with decreasing fracture levels.

In normal bone models, PVMSs in cortical bone and nail constructs (nail body and distal locking screw) were greater than the yield strength at only fracture levels 50 mm below the lesser

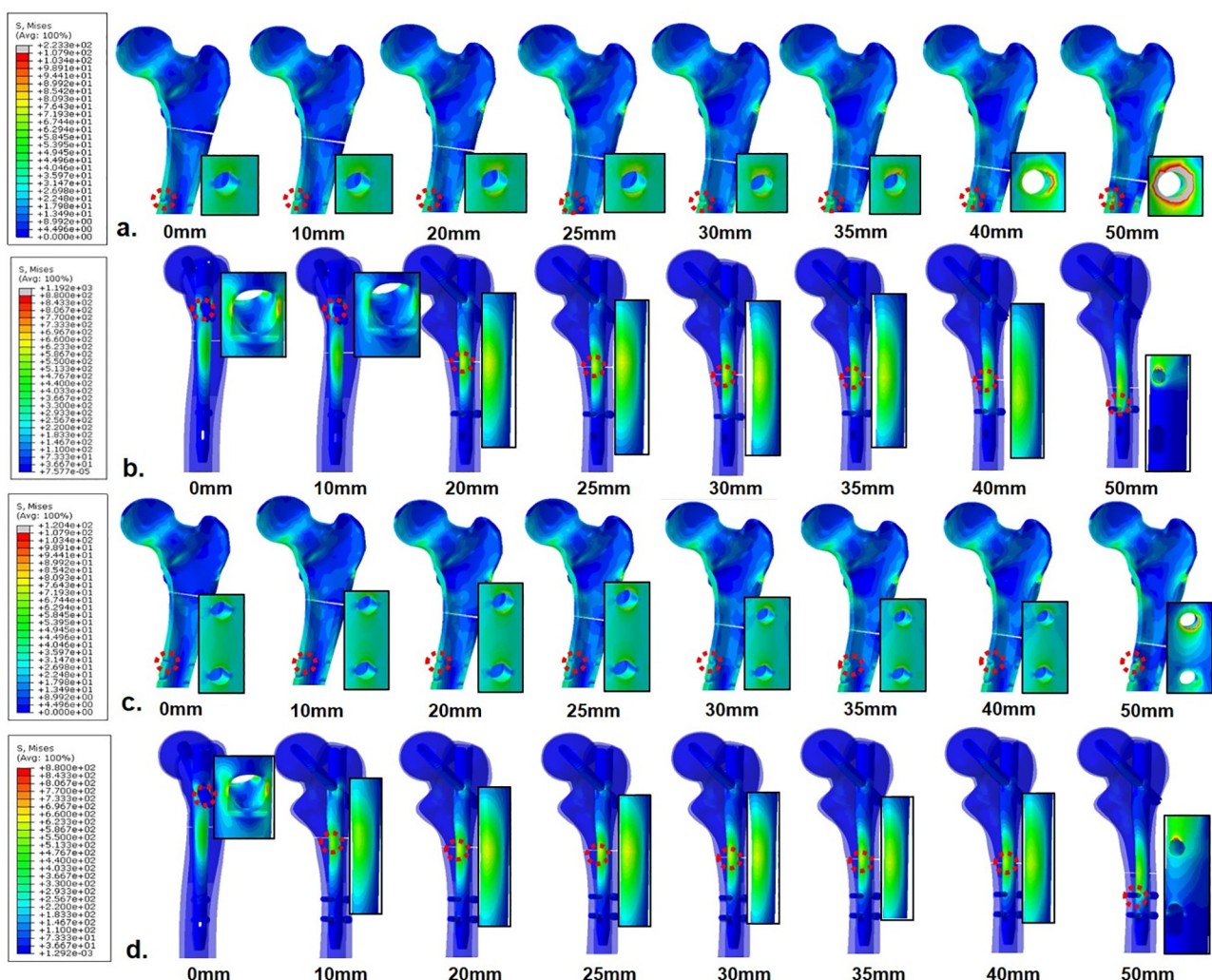

**Fig 3. Stress distribution around the cortical bone and implant of finite element models using 1 (a, b) and 2 (c, d) distal screw fixation in 1 mm fracture gap, osteoporotic bone.** The enlarged image portion represents the point at which the peak stress was observed.

trochanter among FEMs fixed with one distal locking screw. Meanwhile, PVMSs in FEMs fixed with two distal locking screws were less than the yield strength at all fracture levels (S1 Fig). In osteoporotic bone models, PVMSs in the cortical bone at fracture levels 50 mm below the lesser trochanter and the junction of the nail body and the lag screw at 0 mm were greater than the yield strength, regardless of the number of distal locking screws. Meanwhile, in FEMs with one distal locking screw, PVMS in the cortical bone was greater than the yield strength at fracture level 40 mm below the lesser trochanter (Fig 3). Therefore, FEMs fixed with two distal locking screws showed a wider safe range in both normal and osteoporotic bone models.

## Stress distribution in cortical bone and nail constructs in FEMs with 2-mm fracture gap

In normal bone models, PVMSs in cortical bone and nail constructs (nail body and distal locking screw) were greater than the yield strength at fracture levels $\geq 40$ mm and 50 mm below the lesser trochanter, respectively, among FEMs fixed with one distal locking screw. Meanwhile,

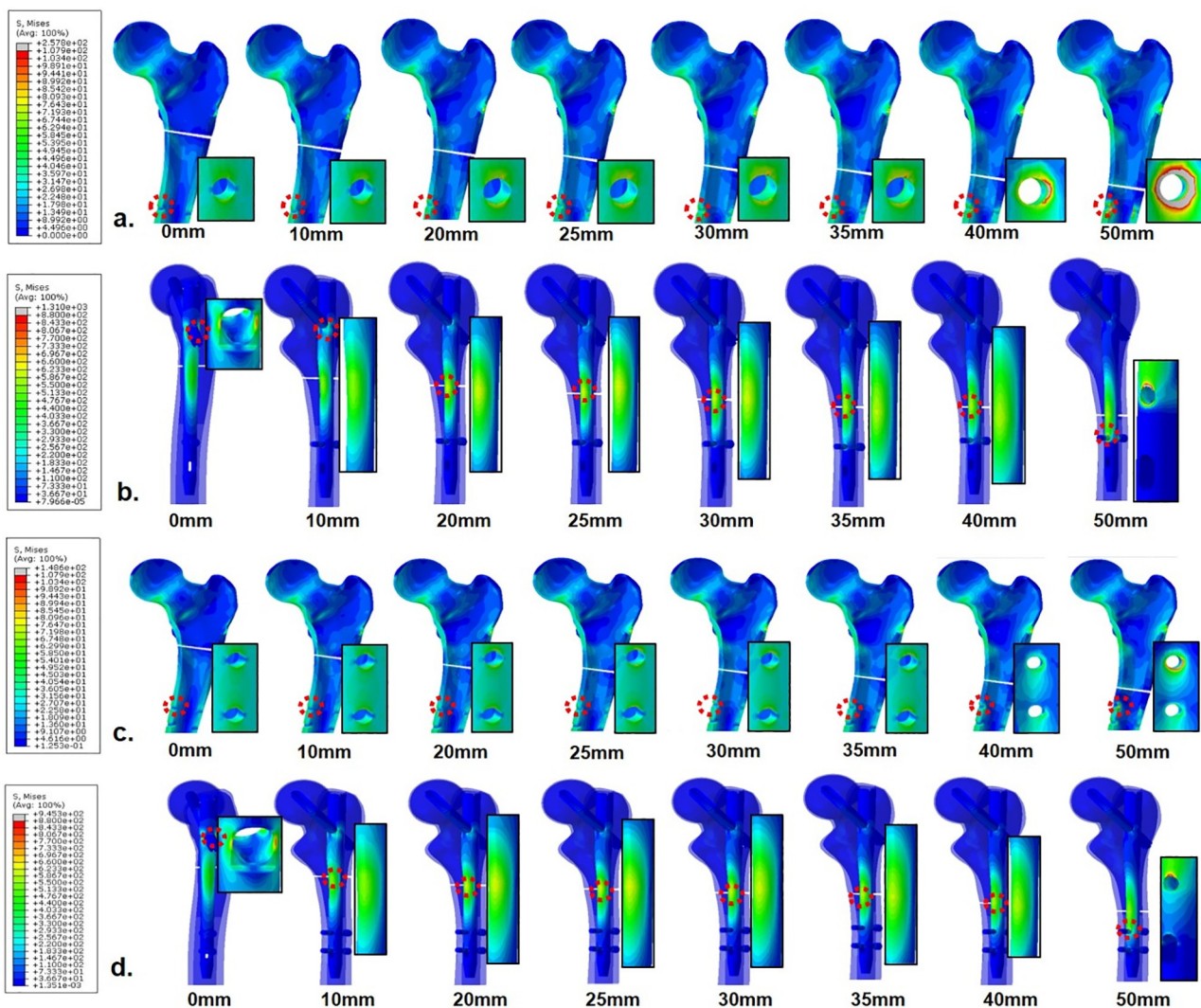

**Fig 4. Stress distribution around the cortical bone and implant of finite element models using 1 (a, b) and 2 (c, d) distal screw fixation in 2 mm fracture gap, osteoporotic bone.** The enlarged image portion represents the point at which the peak stress was observed.

PVMSs in FEMs fixed with two distal locking screws were greater than the yield strength at only 50 mm below the lesser trochanter (S2 Fig). In osteoporotic bone models, PVMSs in the cortical bone at fracture levels 50 mm below the lesser trochanter and the junction of the nail body and the lag screw at 0 and 50 mm were greater than the yield strength, regardless of the number of distal locking screws. Meanwhile, in FEMs with one distal locking screw, PVMS in the cortical bone was greater than the yield strength at fracture levels ≥ 35 mm below the lesser trochanter (Fig 4). Therefore, FEMs fixed with two distal locking screws showed wider safe ranges in both normal and osteoporotic bone models (Table 2).

## Stress distribution in cortical bone and nail constructs in FEMs with 3-mm fracture gap

In normal bone models, PVMSs in the cortical bone and the junction of the nail body and the lag screw at fracture level 50 mms below the lesser trochanter were greater than the yield

**Table 2. Summary of Peak von Mises stress in finite element models according to fracture gap.**

| Fracture gap | Fracture level | Peak Von Mises Stress (MPa) | | | | | | | | | | | | | | | |
|---|---|---|---|---|---|---|---|---|---|---|---|---|---|---|---|---|---|
| | | Cortical bone | | | | Nail body | | | | Lag screw | | | | Distal locking screw | | | |
| | | Normal bone | | Osteoporotic bone | | Normal bone | | Osteoporotic bone | | Normal bone | | Osteoporotic bone | | Normal bone | | Osteoporotic bone | |
| | | 1 screw | 2 screw | 1 screw | 2 screw | 1 screw | 2 screw | 1 screw | 2 screw | 1 screw | 2 screw | 1 screw | 2 screw | 1 screw | 2 screw | 1 screw | 2 screw |
| 1 mm | 0mm | 68 | 65 | 76 | 77 | 789 | 788 | 951* | 949* | 592 | 592 | 761 | 760 | 109 | 131 | 121 | 146 |
| | 10mm | 71 | 67 | 85 | 86 | 542 | 541 | 551 | 548 | 320 | 320 | 411 | 420 | 106 | 128 | 135 | 161 |
| | 20mm | 75 | 71 | 93 | 87 | 625 | 625 | 625 | 621 | 204 | 204 | 299 | 299 | 137 | 169 | 188 | 188 |
| | 25mm | 78 | 74 | 94 | 89 | 640 | 640 | 643 | 639 | 203 | 203 | 249 | 249 | 157 | 185 | 230 | 190 |
| | 30mm | 84 | 79 | 98 | 90 | 614 | 614 | 623 | 622 | 201 | 201 | 208 | 208 | 187 | 190 | 327 | 178 |
| | 35mm | 89 | 81 | 104 | 91 | 606 | 605 | 610 | 605 | 199 | 198 | 203 | 202 | 252 | 187 | 405 | 211 |
| | 40mm | 98 | 81 | 146* | 93 | 582 | 591 | 595 | 593 | 198 | 198 | 205 | 204 | 369 | 225 | 542 | 286 |
| | 50mm | 194* | 101 | 223* | 120* | 1003* | 778 | 1192* | 878 | 188 | 190 | 205 | 205 | 992* | 581 | 1245* | 642 |
| 2 mm | 0mm | 69 | 66 | 76 | 76 | 789 | 788 | 952* | 950* | 593 | 592 | 761 | 761 | 110 | 131 | 121 | 146 |
| | 10mm | 73 | 70 | 89 | 85 | 553 | 552 | 558 | 558 | 320 | 320 | 411 | 411 | 107 | 128 | 139 | 164 |
| | 20mm | 74 | 71 | 95 | 86 | 627 | 616 | 626 | 624 | 204 | 205 | 300 | 300 | 140 | 172 | 192 | 189 |
| | 25mm | 78 | 74 | 95 | 89 | 638 | 637 | 641 | 640 | 203 | 203 | 250 | 250 | 161 | 186 | 243 | 189 |
| | 30mm | 83 | 74 | 102 | 92 | 612 | 612 | 620 | 619 | 201 | 201 | 208 | 209 | 206 | 186 | 317 | 184 |
| | 35mm | 96 | 79 | 109* | 96 | 604 | 603 | 608 | 607 | 198 | 198 | 203 | 202 | 270 | 185 | 415 | 217 |
| | 40mm | 119* | 78 | 157* | 98 | 591 | 589 | 594 | 591 | 198 | 198 | 204 | 203 | 418 | 252 | 594 | 314 |
| | 50mm | 212* | 109* | 258* | 149* | 1106* | 843 | 1310* | 945* | 187 | 189 | 206 | 205 | 1082* | 626 | 1308* | 687 |
| 3 mm | 0mm | 72 | 67 | 81 | 78 | 790 | 789 | 926* | 951* | 593 | 592 | 762 | 763 | 109 | 131 | 121 | 146 |
| | 10mm | 76 | 73 | 87 | 87 | 563 | 563 | 567 | 567 | 321 | 320 | 411 | 411 | 110 | 128 | 144 | 168 |
| | 20mm | 79 | 74 | 92 | 88 | 617 | 626 | 633 | 632 | 206 | 205 | 300 | 300 | 144 | 177 | 207 | 190 |
| | 25mm | 85 | 74 | 97 | 88 | 638 | 637 | 641 | 640 | 204 | 204 | 250 | 250 | 167 | 187 | 253 | 188 |
| | 30mm | 88 | 75 | 111* | 90 | 614 | 614 | 619 | 618 | 202 | 202 | 208 | 208 | 220 | 184 | 337 | 184 |
| | 35mm | 98 | 78 | 135* | 98 | 605 | 604 | 609 | 609 | 198 | 198 | 203 | 202 | 294 | 186 | 442 | 231 |
| | 40mm | 124* | 79 | 159* | 101 | 591 | 598 | 595 | 592 | 197 | 196 | 204 | 204 | 439 | 268 | 641 | 339 |
| | 50mm | 238* | 129* | 273* | 162* | 1215* | 909* | 1435* | 1009* | 185 | 184 | 205 | 205 | 1176* | 670 | 1374* | 609 |

strength, regardless of the number of distal locking screws. PVMS in the cortical bone at fracture level 40 mm was also greater than the yield strength among FEMs fixed with 1 distal locking screw (S3 Fig). In osteoporotic bone models, PVMSs in the cortical bone at 50 mm and the junction of the nail body and the lag screw at 0 and 50 mm were greater than the yield strength, regardless of the number of distal locking screws. However, in FEMs with one distal locking screw, PVMS in the cortical bone was greater than the yield strength at fracture levels $\geq 30$ mm below the lesser trochanter (Fig 5). Therefore, FEMs fixed with two distal locking screws showed a wider safe range in both normal and osteoporotic bone models. Table 2 shows a summary of PVMS in FEMs according to fracture gap.

## Discussion

Several studies have used the finite element method to investigate the biomechanics of subtrochanteric fractures [32–34]. However, our understanding of the optimal management of these fractures are still limited. This study highlighted that the use of a short CMN with two distal locking screws may be a viable option in most transverse subtrochanteric fractures at fracture levels 10 to 40 mm below the lesser trochanter in normal bone and at 10 to 30 mm in osteoporotic bone, under the assumptions of anatomical reduction and fracture gaps $\leq 3$ mm.

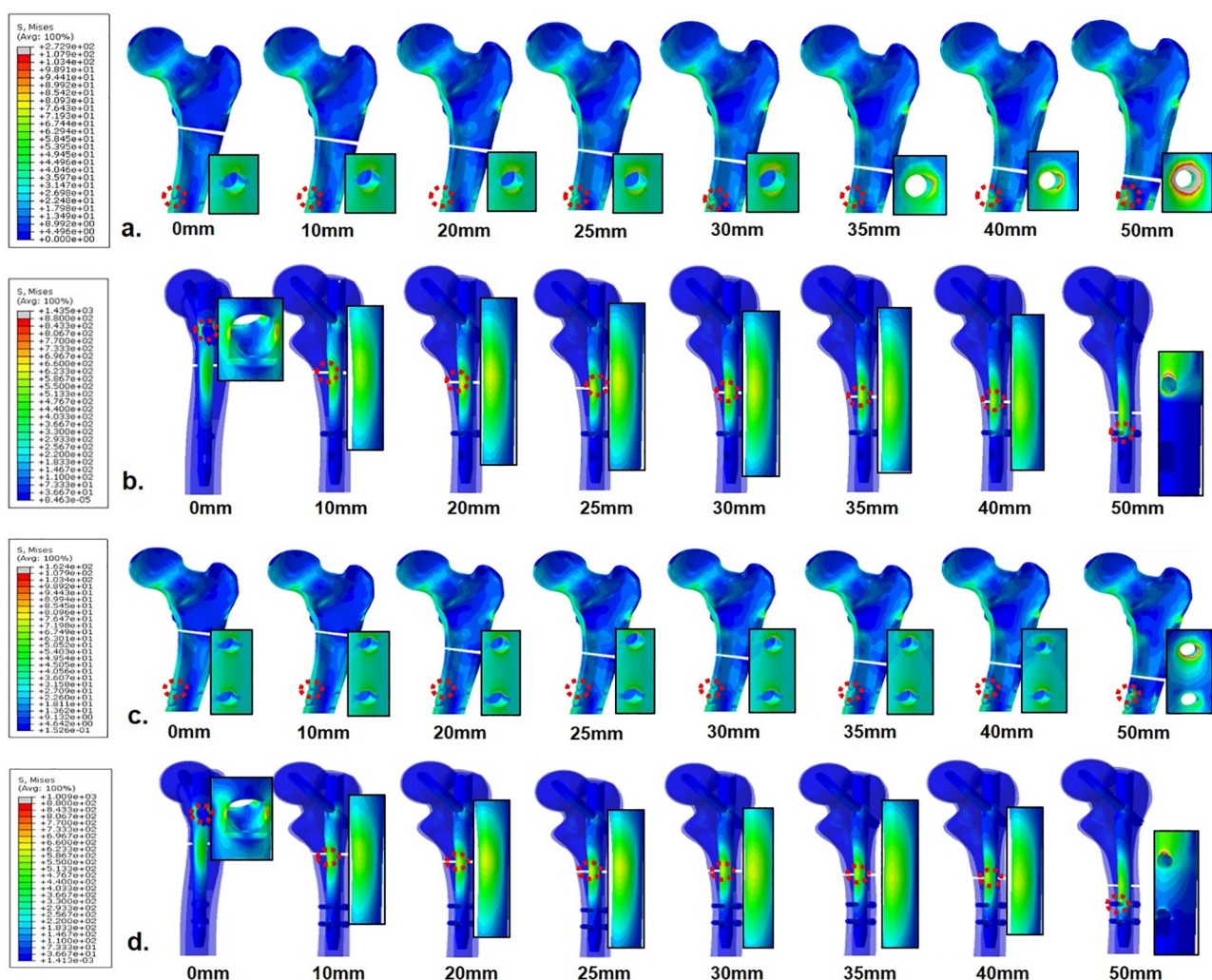

**Fig 5. Stress distribution around the cortical bone and implant of finite element models using 1 (a, b) and 2 (c, d) distal screw fixation in 3 mm fracture gap, osteoporotic bone.** The enlarged image portion represents the point at which the peak stress was observed.

However, the use of one distal locking screw at fracture gaps $\geq$ 2 mm reduces the safe range of fracture level for CMNs and would increase the risk of fixation failure or peri-implant fracture. Therefore, short CMNs with one distal locking screw should be avoided in subtrochanteric fractures even with fracture gaps $\leq$ 3 mm, especially in the osteoporotic bone due to an increase in PVMSs.

CMNs have been widely used in the surgical management of subtrochanteric fractures due to their biomechanical and clinical superiority to extramedullary implants [5,7,8]. Although long nails are generally used for subtrochanteric fractures, short CMNs are also used for high subtrochanteric fractures [35]. However, there are no definite indications or evidences for the use of short CMNs depending on the fracture level and few studies on this issue. The fixation stability in subtrochanteric fractures is of paramount importance because of anatomical geometry of the proximal femur and strong deforming forces in the subtrochanteric region. Several factors including the fracture level, gap, bone quality, nail length, and the number of distal locking screw can affect the fixation stability after nailing in subtrochanteric fractures [36–38].

Therefore, it is difficult to compare and validate the fixation stability clinically under various fixation conditions using short or long nails due to several problems including ethical issues. Besides, there is a lack of evidences on the fixation strength of short CMNs according to fracture level and gap in subtrochanteric fractures and the stresses surrounding those. Therefore, we conducted the current study using FEMs to investigate this issue.

Generally, stress/strain distribution analysis using the finite element methods is widely accepted as a useful technique to evaluate or predict the biomechanical behavior of orthopaedic implants under certain load conditions [33,39,40]. Therefore, this study was conducted to investigate the biomechanics of subtrochanteric fracture fixation using short CMNs using a finite element analysis. In this analysis, we compared the strengths of bone-implant constructs at various fracture levels and gaps using short CMNs with one or two distal locking screws and finally tried to suggest the fracture levels and fixation methods suitable for their use in transverse subtrochanteric fractures in normal and osteoporotic bone models.

All FEMs in this study showed similar stress distribution at PVMS sites around the cortical bone and nail constructs at corresponding fracture levels, regardless of the fracture gap and the number of distal locking screws. PVMS in the cortical bone around the nail was observed at the distal locking screw hole regardless of the fracture level and gap, number of distal locking screws, and bone quality. These values further increased in FEMs using one distal locking screw, and as the fracture level lowered and the fracture gap increased in both normal and osteoporotic bone models. Subsequently, the safe range of the fracture levels for the use of a short CMNs in subtrochanteric fractures was reduced, at which the PVMSs were less than the yield strength. However, PVMSs on this site in osteoporotic FEMs, even fixed with two distal locking screws, showed relatively high values, corresponding to over about 80% of the yield strength at fracture levels 10 to 40 mm below the lesser trochanter, regardless of the fracture gap. Furthermore, PVMSs at the fracture gap $\geq 2$ mm were over approximately 90% of the yield strength at fracture levels 35 and 40 mm. These findings suggest that the distal locking screw hole at the medial or lateral cortex may be a stress-riser, causing peri-implant fracture or fixation failure following subtrochanteric fracture fixation using a short CMNs. Therefore, we believe that two distal locking screws should be used, and the fracture gap should be minimized to $\leq 1$ mm as far as possible when using a short CMN within the safe range of subtrochanteric fractures, especially in osteoporotic bone. Besides, more caution should be taken to prevent refracture around the distal locking screw and protected weight-bearing with walking aids should be maintained to reduce this risk till bony union. Likely, we focused on the risk of fixation failure during the early postoperative period until the bone union. Therefore, we did not consider cyclic loading in this study.

Meanwhile, as the fracture level goes down, the PVMS site on the nail body tended to move downwards from the junction of the nail body and the lag screw to the junction of the nail body and the distal locking screw. These findings are similar to the results of earlier studies that reported breakage of the CMNs at three principal nail points (the junction of the nail and the lag screw, the distal locking screw, and the fracture site), especially as the lag screw hole and the distal locking screw hole of the nail body are weak points due to the narrow area of the nail [41–43]. In our results, PVMSs on the junction of the nail body and the lag screw at fracture level 0 mm measured over 90% of the yield strength even in normal bone, regardless of the fracture gap and the number of distal locking screws. Moreover, PVMSs on the junction of the nail body and the distal locking screw at 50-mm fracture levels measured over 90% of the yield strength in FEMs fixed even with two distal locking screws, regardless of the fracture gap and bone quality. Based on these findings, we believe that short CMNs should not be used at fracture levels 0 and 50 mm even with two distal locking screws at a 1-mm fracture gap in both normal and osteoporotic bone. According to previous literature, short CMNs can be used for

high subtrochanteric fractures but may be contraindicated for low subtrochanteric fractures due to short distance between the fracture site and distal locking screws [35]. However, short CMNs may be contraindicated in high subtrochanteric fractures just below the lesser trochanter even with two distal locking screws, regardless of the fracture gap and bone quality, considering our results.

The finite element study did not represent the true *in vivo* fracture fixation condition. Ideally, the inclusion of all muscles, joint reactions, and the presence of fracture callus would reveal the true nature of the *in vivo* mechanical response [44]. However, this is not straightforward, and simplified loading conditions will continue to be used in experiments and provide biomechanical guidance on the fracture fixation. The FEM in this study was validated by comparison with previously published models. Furthermore, we reproduced various models according to the fracture level and gap, the number of the distal locking screws and bone quality. Therefore, we believe that this experimental study suggests the novel evidence related to the usage of short nails in limited conditions of subtrochanteric fractures although this does not provide the absolute criteria for the usage of short nails in the treatment of subtrochanteric fractures.

There are some limitations to this study. First, the complex physiological force components around the proximal femur were simplified as physiologic loading during activities is more complex, and greater loading can occur in real-life situations. However, only axial loading, which simulated the forces of a 1-legged stance, was considered appropriate for this finite element analysis as protected weight-bearing with walking aids after operation is recommended until bony union is obtained. Second, we conducted this study under the linear static condition only so fatigue fracture was not considered in this study. Fatigue fracture can occur with long-term repeated loading after the fixation for the femoral fractures. However, we focused on the risk of fixation failure or refracture during early postoperative period until the bone union in this study. Therefore, we did not consider fatigue fracture requiring long-term cyclic loading. Third, we could not decide the exact interaction between the nail and the bone. Although the friction coefficient of 0.42 was assumed as the general contact, it is difficult to accept this value for perfect reproducibility as it is difficult to decide the precise interaction at the implant-bone interface. Besides, we did not consider the effect of screw-bone interface modelling in the clinical conditions [45]. However, intramedullary nailing simulated in this study may be less affected by the screw-bone interface than plate-screw fixation and all models would be similarly affected under the same contact condition of distal locking screw-bone interface. Finally, our results would be not universally valid for all subtrochanteric fracture types because we aimed at subtrochanteric transverse fractures, which are consistently reproducible. However, we believe that the use of short CMNs in other patterns of subtrochanteric fractures could be decided based on our results; this is because subtrochanteric fracture patterns are varied, and there is insufficient evidence for the use of short CMNs in these fractures.

A major strength of this study is that, to our knowledge, it is the first finite element analysis study to investigate the stress distribution around short CMNs used in the fixation of subtrochanteric fractures at various fracture levels and gaps and to evaluate the fixation strength according to the number of distal locking screws and bone quality. Finally, this finite element analysis study simulated various situations of 96 FEMs of subtrochanteric fractures fixed using short CMNs according to different fracture levels and gaps, bone quality, and the number of distal locking screws. However, large-cohort clinical studies are needed to verify the results of this study and to determine the viability of short CMNs for subtrochanteric fractures as this is an experimental study using finite element analysis.

## Conclusions

In the current study, two distal locking screws showed a wider safe range than one distal screw when short CMNs were used in subtrochanteric transverse fractures in both normal and osteoporotic bone models under the assumptions of anatomical reduction at fracture gaps ≤ 3 mm. Short CMNs with two distal locking screws may be considered as a suitable option for the fixation of subtrochanteric transverse fractures at fracture levels 10 to 40 mm below the lesser trochanter in normal bone and at 10 to 30 mm in osteoporotic bone under the same assumptions. However, the fracture gap should be reduced to the minimum to lower the risk of refracture and fixation failure, especially in osteoporotic fractures. Accordingly, we carefully suggest that a short CMN may be considered as an available treatment option for subtrochanteric fractures under these conditions. Finally, we believe that our results provide fundamental basic outputs and relative indications for using short CMNs in subtrochanteric fractures.

## Supporting information

**S1 Fig. Stress distribution around the cortical bone and implant of finite element models using 1 (a, b) and 2 (c, d) distal screws fixation in 1 mm fracture gap, normal bone.** The enlarged image portion represents the point at which the peak stress was observed.
(TIF)

**S2 Fig. Stress distribution around the cortical bone and implant of finite element models using 1 (a, b) and 2 (c, d) distal screws fixation in 2 mm fracture gap, normal bone.** The enlarged image portion represents the point at which the peak stress was observed.
(TIF)

**S3 Fig. Stress distribution around the cortical bone and implant of finite element models using 1 (a, b) and 2 (c, d) distal screws fixation in 3 mm fracture gap, normal bone.** The enlarged image portion represents the point at which the peak stress was observed.
(TIF)

## Author Contributions

**Conceptualization:** Dae-Kyung Kwak, Je-Hyun Yoo.

**Data curation:** Won-Hyeon Kim, Seunghun Lee.

**Formal analysis:** Sung-Jae Lee.

**Investigation:** Seunghun Lee.

**Methodology:** Sun-Hee Bang, Sung-Jae Lee.

**Software:** Sun-Hee Bang.

**Supervision:** Je-Hyun Yoo.

**Validation:** Won-Hyeon Kim.

**Writing – original draft:** Dae-Kyung Kwak.

**Writing – review & editing:** Je-Hyun Yoo.

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
