## [Decision Letter · Decision Letter 0]

21 Apr 2021

PONE-D-21-07391

Biomechanics of subtrochanteric fracture fixation using short cephalomedullary nails: a Finite element analysis

PLOS ONE

Dear Dr. Yoo,

Thank you for submitting your manuscript to PLOS ONE. After careful consideration, we feel that it has merit but does not fully meet PLOS ONE’s publication criteria as it currently stands. Therefore, we invite you to submit a revised version of the manuscript that addresses the points raised during the review process.

We look forward to receiving your revised manuscript.

Kind regards,

Antonio Riveiro Rodríguez, PhD

Academic Editor

PLOS ONE

https://www.injuryjournal.com/article/S0020-1383(19)30476-0/fulltext

In your revision ensure you cite all your sources (including your own works), and quote or rephrase any duplicated text outside the methods section. Further consideration is dependent on these concerns being addressed.

Reviewers' comments:

Reviewer's Responses to Questions

**Comments to the Author**

1. Is the manuscript technically sound, and do the data support the conclusions?

Reviewer #1: Yes

Reviewer #2: Yes

Reviewer #3: Yes

2. Has the statistical analysis been performed appropriately and rigorously? 

Reviewer #1: N/A

Reviewer #2: N/A

Reviewer #3: Yes

3. Have the authors made all data underlying the findings in their manuscript fully available?

Reviewer #1: Yes

Reviewer #2: Yes

Reviewer #3: Yes

4. Is the manuscript presented in an intelligible fashion and written in standard English?

Reviewer #1: Yes

Reviewer #2: Yes

Reviewer #3: Yes

5. Review Comments to the Author

Reviewer #1: This paper presents a comparison of the stress distribution in the implant and surrounding cortical bone that appear after implantation of short cephalomedullary nails to fix subtrochanteric fracture, when changing some variables. These are: the distance of the fracture from the lesser trochanter (0 to 50mm), the fracture gap (1,2 and 3 mm), using 1 or 2 distal screws, and considering normal or osteoporotic bone. This makes a total of 92 FE analyses that are compared in terms of the value and location of the peak von Mises stress.

There are not many works on this specific application. Also, the conclusions are clinically useful, despite the limitations of the models used. The paper is well-written, although too repetitive, clear and well-focused. Also, the methodological approach is sound and sufficiently described.

On the side of the cons, there is nothing new in the methodology. The simplest approach possible (one single reference load, a very simplified and inaccurate description of the femoral supporting boundary conditions, linearly elastic and isotropic properties both for cortical and cancellous bone, a rough reduction of elastic modulus for the osteoporotic bone, simplified contact conditions) has been used. Therefore, the utility of the paper is essentially clinical. Despite the above limitations, and for comparative purpose, as it is the one of this paper, the results are useful and comparable, so the clinical conclusions are reasonably well supported by the results.

The Discussion section is mainly a summary and repetition of the results obtained, and a comparison between them, which is right, but I would have expected a more complete comparison between the conclusions and what is known in clinical practice, that is, I miss a more clinical discussion on the results.

The limitations of the paper are mostly described in the Discussion, except for the effect of boundary conditions, although, since they are distally applied, I do not expect a big influence on the results on the proximal part, especially when talking about comparative results. However, I would like to see the opinion of the authors regarding the possible effect on such results of the limitations described.

As an overall comment, I do not think that supporting such conclusions requires to include in the paper the results of all 96 models analyzed in three tables and in 12 figures (6 of them in supplementary material). It would be sufficient to add only one table with the most representative results and only one figure to illustrate the type of results obtained. With all this, may be a short note would be sufficient.

As a summary, the paper is original and there is a clear clinical contribution in a very specific problem, but, since there is no new method and the main conclusions are clinical, and correspond to a very specific type of surgery, a clinical journal in traumatology or orthopedic surgery would be more appropriate. There, a more clinical approach would be expected, which would enrich the paper.

Reviewer #2: This paper presents the results of the FE-based analysis of the stresses distribution around nails and cortical bones in subtrochanteric fracture models.

The topic is worth investigation. The subject of the article is well suited to PLOSONE journal.

The quality of English communication is very good. This manuscript is very well written, with clear structure and careful explanations throughout, enabling others to replicate these techniques if desired.

In my opinion this manuscript presents innovative methods which may find interest in practical applications.

Globally, each scientific point is well explained. So I feel that this manuscript is acceptable for publication after minor revision.

Abbreviations LT and PVMS in the abstract should be explained for non-specialist readers.

Has mesh sensitivity analysis been performed? If not, how was the size of the elements determined to obtain reliable results.

The quality of the figures in the reviewed file is very poor. Especially the legends of the von Mises stress in Figs. 3-8 are blurry.

Reviewer #3: The paper is suffering from poor writing. the comments are as follow:

1-Line 48, the authors mentioned that 96 models were tested. Please changes to 96 finite element models were simulated.

2- The quality of all figures are low. Please improve it. So I will take my time to explain how can you get a high quality image from ABAQUS: cTRL P///file///increase the quality to 4090.

3- The conclusion is too short.

6. PLOS authors have the option to publish the peer review history of their article (what does this mean?). If published, this will include your full peer review and any attached files.

Reviewer #1: No

Reviewer #2: No

Reviewer #3: **Yes: **Rohola Rahnavard

---

## [Author Response · Author response to Decision Letter 0]

5 Jun 2021

5. Review Comments to the Author

Reviewer #1: This paper presents a comparison of the stress distribution in the implant and surrounding cortical bone that appear after implantation of short cephalomedullary nails to fix subtrochanteric fracture, when changing some variables. These are: the distance of the fracture from the lesser trochanter (0 to 50mm), the fracture gap (1,2 and 3 mm), using 1 or 2 distal screws, and considering normal or osteoporotic bone. This makes a total of 92 FE analyses that are compared in terms of the value and location of the peak von Mises stress.

There are not many works on this specific application. Also, the conclusions are clinically useful, despite the limitations of the models used. The paper is well-written, although too repetitive, clear and well-focused. Also, the methodological approach is sound and sufficiently described.

On the side of the cons, there is nothing new in the methodology. The simplest approach possible (one single reference load, a very simplified and inaccurate description of the femoral supporting boundary conditions, linearly elastic and isotropic properties both for cortical and cancellous bone, a rough reduction of elastic modulus for the osteoporotic bone, simplified contact conditions) has been used. Therefore, the utility of the paper is essentially clinical. Despite the above limitations, and for comparative purpose, as it is the one of this paper, the results are useful and comparable, so the clinical conclusions are reasonably well supported by the results.

The Discussion section is mainly a summary and repetition of the results obtained, and a comparison between them, which is right, but I would have expected a more complete comparison between the conclusions and what is known in clinical practice, that is, I miss a more clinical discussion on the results.

-> Thank you for your invaluable comments. 

We have added more clinical discussion in ‘Discussion’ section. However, as mentioned in our study, comparison of the fixation stability and possibility of the fixation failure among the different fixation methods is difficult in clinical conditions, especially in subtrochanteric fractures. Accordingly, there is a lack of clinical comparison study of the fixation stability or fixation failure in subtrochanteric fractures only. Therefore, we believe that this FEA study suggests the novel evidence related to the usage of short nails in limited conditions of subtrochanteric fractures even though this is an experimental study.

The limitations of the paper are mostly described in the Discussion, except for the effect of boundary conditions, although, since they are distally applied, I do not expect a big influence on the results on the proximal part, especially when talking about comparative results. However, I would like to see the opinion of the authors regarding the possible effect on such results of the limitations described.

-> As you know, we have conducted this study using finite element models fully fixed at the distal end (Fig. 2). This is a widely used model not only for this study (Reference 1, 2, and 3), and we have already published some studies using this model (Reference 4, 5). To implement the walking status and observe the physiologic deformity of the femur, it is necessary to fix the distal part of the model. As mentioned in the ‘Discussion’ section, it could not perfectly represent the true in vivo fracture fixation condition. However, stress/strain distribution analysis using the finite element methods is widely accepted as a useful technique. Furthermore, the friction coefficient and loading condition applied as the boundary condition were also based on the previously verified reference. Therefore, we believe that our study condition sufficiently reproduced the clinical conditions.

Reference

1. Eberle S, Gerber C, von Oldenburg G, Hungerer S, Augat P. Type of hip fracture determines load share in intramedullary osteosynthesis. Clin Orthop Relat Res. 2009

2. Lu J, Wang QY, Sheng JG, Guo SC, Tao SC. A 3D-printed, personalized, biomechanics-specific beta-tricalcium phosphate bioceramic rod system: personalized treatment strategy for patients with femoral shaft non-union based on finite element analysis. BMC Musculoskelet Disord. 2020

3. Panagopoulos A, Kyriakopoulos G, Anastopoulos G, Megas P, Kourkoulis SK. Design of Improved Intertrochanteric Fracture Treatment (DRIFT) Study: Protocol for Biomechanical Testing and Finite Element Analysis of Stable and Unstable Intertrochanteric Fractures Treated With Intramedullary Nailing or Dynamic Compression Screw. JMIR Res Protoc. 2019

4. Kwak DK, Kim WH, Lee SJ, Rhyu SH, Jang CY, Yoo JH. Biomechanical comparison of three different intramedullary nails for fixation of unstable basicervical intertrochanteric fractures of the proximal femur: Experimental studies. Bio Med Res Int 2018.

5. Jang CY, Bang SH, Kim WH, Lee SJ, Lee HM, Kwak DK, et al. Effect of fracture levels on the strength of bone-implant constructs in subtrochanteric fracture models fixed using short cephalomedullary nails: A finite element analysis. Injury 2019.

As an overall comment, I do not think that supporting such conclusions requires to include in the paper the results of all 96 models analyzed in three tables and in 12 figures (6 of them in supplementary material). It would be sufficient to add only one table with the most representative results and only one figure to illustrate the type of results obtained. With all this, may be a short note would be sufficient.

-> Thank you for your comments. We have presented all the results in one table and 3 figures as per your comment. The rest of them have moved to the supplementary material. 

Reviewer #2: This paper presents the results of the FE-based analysis of the stresses distribution around nails and cortical bones in subtrochanteric fracture models.

The topic is worth investigation. The subject of the article is well suited to PLOSONE journal.

The quality of English communication is very good. This manuscript is very well written, with clear structure and careful explanations throughout, enabling others to replicate these techniques if desired.

In my opinion this manuscript presents innovative methods which may find interest in practical applications.

Globally, each scientific point is well explained. So, I feel that this manuscript is acceptable for publication after minor revision.

Abbreviations LT and PVMS in the abstract should be explained for non-specialist readers.

 -> So sorry, it was our mistake. We have corrected the abbreviations ‘LT’ and ‘PVMS’ in the abstract. 

Has mesh sensitivity analysis been performed? If not, how was the size of the elements determined to obtain reliable results.

-> Eight-noded hexahedral elements and four-noded tetrahedral elements were created by using the automatic solid and mesh generation program (ABAQUS) to build up the mesh of the fractured femur model and the Gamma3 CMN. These elements enabled the reproduction of the different material properties and maintained contact conditions in the fracture plane. Since there were too many models considered in this study, mesh convergence was not carried out on all models. Instead, the mesh types and sizes were maintained constant for all parts of each model to minimize the possibility of mesh-induced discrepancies.

The quality of the figures in the reviewed file is very poor. Especially the legends of the von Mises stress in Figs. 3-8 are blurry.

-> So sorry. We have changed high quality of the figures in the revised manuscript.

Reviewer #3: The paper is suffering from poor writing. the comments are as follow:

1-Line 48, the authors mentioned that 96 models were tested. Please changes to 96 finite element models were simulated.

-> We have corrected that sentence as per your comment.

2- The quality of all figures is low. Please improve it. So, I will take my time to explain how can you get a high quality image from ABAQUS: cTRL P///file///increase the quality to 4090.

-> So sorry. We have changed high quality of the figures in the revised manuscript.

3- The conclusion is too short.

-> Thank you for your comment. We have added the conclusion as per your comment.

---

## [Decision Letter · Decision Letter 1]

15 Jun 2021

Biomechanics of subtrochanteric fracture fixation using short cephalomedullary nails: a Finite element analysis

PONE-D-21-07391R1

Dear Dr. Yoo,

We’re pleased to inform you that your manuscript has been judged scientifically suitable for publication and will be formally accepted for publication once it meets all outstanding technical requirements.

Kind regards,

Antonio Riveiro Rodríguez, PhD

Academic Editor

PLOS ONE

Additional Editor Comments (optional):

Reviewers' comments:

Reviewer's Responses to Questions

**Comments to the Author**

1. If the authors have adequately addressed your comments raised in a previous round of review and you feel that this manuscript is now acceptable for publication, you may indicate that here to bypass the “Comments to the Author” section, enter your conflict of interest statement in the “Confidential to Editor” section, and submit your "Accept" recommendation.

Reviewer #1: All comments have been addressed

Reviewer #2: All comments have been addressed

2. Is the manuscript technically sound, and do the data support the conclusions?

Reviewer #1: Yes

Reviewer #2: Yes

3. Has the statistical analysis been performed appropriately and rigorously? 

Reviewer #1: N/A

Reviewer #2: N/A

4. Have the authors made all data underlying the findings in their manuscript fully available?

Reviewer #1: Yes

Reviewer #2: Yes

5. Is the manuscript presented in an intelligible fashion and written in standard English?

Reviewer #1: Yes

Reviewer #2: Yes

6. Review Comments to the Author

Reviewer #1: (No Response)

Reviewer #2: The authors have addressed all the comments and suggestions I made in the first review. The quality of the article has been significantly improved. I feel that this manuscript is now acceptable for publication.

7. PLOS authors have the option to publish the peer review history of their article (what does this mean?). If published, this will include your full peer review and any attached files.

Reviewer #1: No

Reviewer #2: No

---

## [Editor Report · Acceptance letter]

17 Jun 2021

PONE-D-21-07391R1 

Biomechanics of subtrochanteric fracture fixation using short cephalomedullary nails: a Finite element analysis 

Dear Dr. Yoo:

I'm pleased to inform you that your manuscript has been deemed suitable for publication in PLOS ONE. Congratulations! Your manuscript is now with our production department. 

Kind regards, 

on behalf of

Dr. Antonio Riveiro Rodríguez 

Academic Editor

PLOS ONE